# Age Specific Risks of Uterine Cancer in Type 2 Diabetes and Associated Comorbidities in Taiwan

**DOI:** 10.3390/cancers14194912

**Published:** 2022-10-07

**Authors:** Hui-Shan Liu, Chin-Der Chen, Chung-Chen Lee, Yong-Chen Chen, Wen-Fang Cheng

**Affiliations:** 1Department of Obstetrics and Gynecology, Fu Jen Catholic University Hospital, Fu Jen Catholic University, New Taipei City 242, Taiwan; 2Data Science Center, College of Medicine, Fu Jen Catholic University, New Taipei City 242, Taiwan; 3Master’s Program of Big Data in Biomedicine, College of Medicine, Fu Jen Catholic University, New Taipei City 242, Taiwan; 4Department of Obstetrics and Gynecology, National Taiwan University Hospital, National Taiwan University, Taipei 100, Taiwan

**Keywords:** uterine cancer, diabetes mellitus (DM), polycystic ovarian syndrome (PCOS), hyperlipidemia (HPL), obesity, hormone replacement therapy (HRT), statins

## Abstract

**Simple Summary:**

The incidence of uterine cancer is increasing worldwide in recent decades, especially in the young age population. In parallel, a trend of early onset of diabetes mellitus (DM) and obesity was also observed in the young population. Since DM and obesity are known risk factors of uterine cancer, we proposed to investigate the age-specific risks of uterine cancer in subjects with DM and related metabolic comorbidities. In this nationwide population-based study, we found the risk of uterine cancer in DM subjects was significantly higher in the younger age groups, especially in the age group 30–39 years. Obesity, polycystic ovarian syndrome, hyperlipidemia, statin use, and hormone replacement therapy were also significantly associated with uterine cancer in subjects younger than 50 years of age. Therefore, subjects with DM, especially younger women with respective comorbidities, should be recommended to have appropriate uterine cancer screenings for early detection.

**Abstract:**

Introduction: The global incidence of uterine cancer has increased substantially in recent decades. We evaluated if the trend of increasing prevalence of diabetes mellitus (DM) and obesity are attributed to the development of uterine cancer. Methods: Using data derived from the National Health Insurance database and Taiwan Cancer Registry, multivariate Cox proportional hazards regression models were adapted to analyze the risk factors of uterine cancer with potential confounding variables. Results: There were a total of 5,104,242 women aged 30–70 years enrolled in the study and 147,772 of them were diagnosed with DM during 2005–2007. In a total of 11 years of follow-up, 14,398 subjects were diagnosed with uterine cancer. An elevated risk of uterine cancer was observed in women with DM of all ages (HR 1.66, 95% CI 1.53–1.81, *p* < 0.0001). The effect of DM was highest at age 30–39 years (RR 3.05, 95% CI 2.35–3.96, *p* < 0.0001). In the group of <50 years old, DM patients had at least a twofold higher risk of developing uterine cancer (HR 2.39, 95% CI 2.09–2.74, *p* < 0.0001). Subjects among all ages diagnosed with polycystic ovary syndrome (PCOS) (HR 2.91, 95% CI 2.47–3.42, *p* < 0.0001), obesity (HR 2.13, 95% CI 1.88–2.41, *p* < 0.0001), and those undergoing hormone replacement therapy (HRT) (HR 1.60, 95% CI 1.33–1.93, *p* < 0.0001) were also positively associated with uterine cancer. Positive associations of hyperlipidemia (HR 1.33, 95% CI 1.22–1.46, *p* < 0.0001) and statin use (HR 1.27, 95% CI 1.12–1.44, *p* = 0.0002) on uterine cancer were only observed in subjects <50 years. On the contrary, hyperlipidemia was negatively associated with uterine cancer in subjects ≥50 years (HR 0.91, 95% CI: 0.84–0.98, *p* = 0.0122). Conclusions: DM is in general the most important risk factor for uterine cancer, especially in premenopausal women. Obesity, PCOS, HPL, statin use, and HRT were also associated with uterine cancer in subjects younger than 50 years. Premenopausal women with DM and respective comorbidities should be aware of the development of uterine cancer.

## 1. Introduction

In the United States, uterine cancer ranked as the fourth most common cancer and the seventh leading cause of cancer death in women [1], whose incidence and mortality are increasing in contrast to other cancers [2]. Uterine cancer is divided into uterine carcinoma and sarcoma based on histopathological characteristics. Uterine carcinoma accounts for the majority of uterine cancer, while uterine sarcoma accounts for only 3–7% of all uterine cancer [3]. Uterine carcinoma, also known as endometrial cancer (EC) in some literature, has been classified into type 1 and type 2 based on histopathological characteristics and clinical outcomes. Among type 1 EC, endometrioid adenocarcinoma represents the most common cell type, and it is featured with a relatively favorable clinical outcome. Endometrioid adenocarcinoma is predisposed by prolonged or excessive estrogen exposure, such as the use of unopposed estrogen during hormone-replacement therapy, estrogen-producing tumors (e.g., ovarian granulosa-cell tumors) or associated with obesity [4]. Instead, type 2 EC, including clear cell carcinoma, serous carcinoma, and carcinosarcoma, is typically poorly differentiated with a higher risk of metastasis and a poor overall outcome [5]. In Taiwan, uterine cancer, especially endometrioid adenocarcinoma, was the most rapidly increasing malignancy at a younger age in the past 30 years [6]. Several risk factors have been reported to be associated with an increased risk of EC development, including diabetes, obesity, old age, family history of Lynch syndrome, early menarche, nulliparity, prolonged or excessive unopposed estrogen stimulation, and late menopause. Among them, type 2 diabetes mellitus (DM) is significantly associated with a twofold higher risk of developing EC compared to individuals without DM. Moreover, DM itself has been considered as an independent risk factor of EC [7,8].

In Taiwan, the incidence and prevalence of DM are steadily increasing, especially in the young population [9]. Similarly, the incidence of obesity in the young population gradually increases in parallel to the trend of DM incidence as well [10]. Collectively, the trend of early onset of DM and obesity as well as increased incidence of EC in younger population were observed.

DM and obesity both belong to the spectrum of metabolic syndrome, which is a collection of several clinical conditions, including insulin resistance, hyperlipidemia (HPL), and hypertension. Among these, HPL is also associated with an increased risk of cancer [11]; however, the association between HPL and EC is limited. A positive association between serum triglyceride level and risk of EC has been reported, but not observed in total cholesterol, low-density lipoprotein cholesterol (LDL) and high-density lipoprotein cholesterol (HDL) [12]. 

Previously, Chen et al. reported that DM significantly increases the risk of EC in all age, and the highest relative risk was observed in women younger than 50 years of age [13]. However, this finding should be interpreted with caution because several potential confounders, such as obesity and polycystic ovarian syndrome (PCOS), were not adjusted accordingly. PCOS is a common disease in young women that is associated with EC development [14]. Additionally, the trend of early onset of obesity might also confound their analysis results. 

Therefore, we proposed to utilize a representative nationwide database, the National Health Insurance (NHI) database, and the Taiwan Cancer Registry (TCR), to investigate the age-specific risks of uterine cancer in subjects with diabetes and related metabolic comorbidities.

## 2. Materials and Methods

### 2.1. Study Design and Population

The information of the subjects was obtained from the National Health Insurance Database in Taiwan. This single-payer National Health Insurance Program was launched in 1995 and the entire population in Taiwan, nearly 23 million subjects, were enrolled in this program. Women aged between 30–70 years old from 1 January 2005 to 31 December 2015 were in this analysis. Women younger than 30 years were excluded to ensure that most of the DM were type 2, since the onset of type 1 DM occurs more frequently in people under 30 years of age. Women aged older than 70 years were excluded since most uterine cancers were diagnosed before 70. Additional exclusion criteria included: (1) subjects diagnosed with DM before 2005, (2) subjects diagnosed with any cancer before 2005, (3) women with a medical record of hysterectomy in 2004, (4) uterine cancer diagnosed prior to the diagnosis of DM, and (5) hysterectomy prior to the diagnosis of DM. 

We use the International Classification of Diseases, Ninth Revision, Clinical Modification (ICD-9-CM) to code the diagnosis of uterine cancer, DM, obesity, PCOS, and HPL of the subjects enrolled (Appendix A). In Taiwan, the diagnosis of DM was based on the American Diabetes Association for diabetes mellitus [15,16]. In order to increase the diagnostic validity, the diagnosis of DM was made only when three or more consecutive outpatient visits in a year during follow-up between 2005 to 2007 in this study. 

Vital status during the follow-up period and pathological subtypes of uterine cancer in the enrolled subjects were determined by data link with the healthy profiles of the National Cancer Registry and the National Death Certification System in Taiwan, respectively. Taiwan Cancer Registry (TCR) is a medical system that contains information for epidemiology, diagnosis, and treatment summary of newly diagnosed cancer cases in Taiwan since 1979 [17]. The completeness of the TCR database was 98.4% in 2016 [17]. The cellular or pathological confirmation rate of cancer diagnosis was up to 99.68% according to the Taiwan Cancer Registry Annual Report in 2019 (https://www.hpa.gov.tw/Pages/Detail.aspx?nodeid=269&pid=14913, accessed on 13 January 2022). All subjects enrolled in this study were regularly followed up either until the end of 2015 or until the date of censoring. The date of censoring was (1) the date of diagnosis of uterine cancer, (2) the mortality time, or (3) the date of hysterectomy. This project was approved by the Ethics Committee and the Institutional Review Board of Fu Jen Catholic University (IRB No. C107021), New Taipei City, Taiwan.

### 2.2. Uterine Cancer

Histopathological subtypes of uterine cancers were classified as carcinoma and sarcoma. Uterine carcinomas include endometrioid adenocarcinoma, serous adenocarcinoma, clear cell adenocarcinoma and carcinosarcoma. Uterine sarcomas were composed of low grade endometrial stromal sarcoma (LGESS), endometrial stromal sarcoma (ESS), leiomyosarcoma (LMS) and adenosarcoma [6]. Subjects diagnosed with unclassified uterine cancer were categorized as others. Histopathological classification of the subtypes of uterine cancers was based on the World Health Organization classification of tumors listed in Appendix A.

### 2.3. Comorbidity

Comorbidities, including PCOS, obesity, and HPL, were determined based on medical records of corresponding ICD codes between 2005 and 2007 (Appendix A). Obesity is diagnosed when BMI > 27 kg/m^2^ using the criteria defined by the Department of Health in Taiwan. Hormone replacement therapy (HRT) was determined according to the Anatomical Therapeutic Chemical (ATC) classification system, and prescription data were retrieved from the NHI database (Appendix A). Either treatment with estrogen-only (ATC code: G03C) or estrogen-progesterone combination (ATC code: G03F) were categorized as HRT [18]. Patients who had ever been prescribed with statins, no matter how long was the treatment duration, were categorized into the group of statins. 

### 2.4. Statistical Analysis

We utilized Cox proportional hazard regression models to assess the independent effects of DM, PCOS, obesity, HPL, use of HRT, and use of statins on the development of uterine cancer. Potential confounding factors including age, DM, PCOS, obesity, HPL and use of HRT and statins were adjusted accordingly. We further divided age groups into <50 years (premenopausal) and ≥50 years (postmenopausal) groups to see the age-specific risk of developing uterine cancer in subjects with DM, PCOS, obesity, HPL and under HRT, and use of statins in both age groups. Moreover, we performed a stratified analysis for each given comorbidity in subjects with or without DM to clarify the independent risk of each comorbidity. All significance tests were two-tailed, and the value of α was 0.05. All statistical analyses were performed using the Statistical Analysis System (SAS version 9.4).

## 3. Results

### 3.1. Basic Characteristics

There was a total of 11,164, 833 women seeking medical care in 2005 to 2007 based on the NHI database, and 6,274,517 subjects were excluded after applying exclusion criteria (Figure 1). Subsequently, 5,104,242 subjects were enrolled in the study after linking with TCR. The total follow-up was 52,787,303.3 person years. The basic demographics including age, comorbidities, and use of statins of the population are outlined in Table 1. There was a total of 147,772 subjects diagnosed with DM between January 2005 and December 2007, and the overall incidence rate of DM was 2.9% (Table 1). The mean age was 54.55 ± 9.2 years in the DM group and 45.13 ± 10.19 years in the non-DM group, respectively (*p* < 0.0001). It indicates that subjects diagnosed with DM were older than those in the non-DM group. 

As expected, more metabolic comorbidities were observed in the DM group, especially obesity and HPL. HPL was present in 39.42% of subjects in the DM group, but only 8.27% in the non-DM group (*p* < 0.0001). Regarding obesity, it was 3.09% and 0.69% in the DM and non-DM groups, respectively (*p* < 0.0001). Statins were also prescribed more frequently in the DM group than in the non-DM group (41.51% vs. 5.31%, *p* < 0.0001). In contrast, HRT was prescribed more frequently in the non-DM group (0.48% vs. 0.44%, *p* = 0.0465). Rate of PCOS was similar between these two groups (0.57% vs. 0.54% in DM and non-DM groups).

### 3.2. Basic Characteristics of Subjects with Uterine Cancer

The basic characteristics of the subjects of uterine cancer were summarized in Table 2A. A total of 14,398 subjects were diagnosed with uterine cancer with an overall incidence rate of 27.28/100,000 person-years. In the DM group, 678 of 147,772 (0.46%), while 13,720 of 4,956,470 (0.28%) in the non-DM group developed uterine cancer. The incidence of uterine cancer increased significantly in the DM group (*p* < 0.0001). Similar to general cohort (Table 1), in this sub-cohort of uterine cancer, DM subjects were significantly older, with more obesity, HPL, and use of statins. However, there was no statistical significance of using HRT between the DM and non-DM groups (Table 2), 

### 3.3. Subtypes of Uterine Cancer and DM 

With respect to the cancer subtypes, there was no difference in the subtypes, carcinoma, or sarcoma of uterine cancer between non-DM and DM groups (Table 2B). The main subtype of uterine cancer was endometrioid adenocarcinoma in both groups (non-DM 55.88% vs. DM 55.60%). Type 2 carcinoma (clear cell, serous and carcinosarcoma) accounted for 3.72% and 4.28% of uterine cancer in the non-DM and DM groups, respectively. There was no difference in the incidence of overall sarcoma or any subtype of sarcoma between the groups. 

The accumulated probability of uterine cancer in subjects with DM and without DM during follow-up years is shown in Figure 2. Subjects with DM had a higher accumulated probability of uterine cancer (The log rank test, *p* < 0.0001).

### 3.4. Risk Factors Correlated with the Development of Uterine Cancer

In the multivariate Cox proportional hazard regression model, with or without adjustment, age was positively associated with uterine cancer (Table 3). The 50–59 age group has the highest risk (HR 2.87, 95% CI 2.72–3.01, *p* < 0.0001) of developing uterine cancer followed by the 40–49 age group (HR 2.53, 95% CI 2.41–2.65, *p* < 0.0001) and 60–69 (HR 1.69, 95% CI 1.58–1.81, *p* < 0.0001). Among DM and comorbidities, PCOS has the highest HR among all potential risk factors (HR 2.91, 95% CI 2.47–3.42, *p* < 0.0001). Obesity and DM were significantly associated with risks of developing uterine cancer (obesity, HR 2.13, 95% CI 1.88–2.41, *p* < 0.0001; DM, HR 1.66, 95% CI 1.53–1.81, *p* < 0.0001). Furthermore, subjects under HRT were also at risk of developing uterine cancer (HR 1.6, 95% CI 1.33–1.93, *p* < 0.0001). HPL and use of statins were not correlated with the development of uterine cancer. 

Our findings indicated that subjects of specific age, diagnosis of PCOS, obesity, DM, and HRT were at an increased risk of developing uterine cancer. 

### 3.5. Relative Risk (RR) of DM and Uterine Cancer

To further explore the effect of DM on the development of uterine cancer, we examined the RR of developing uterine cancer in subjects with and without DM in a variety of comorbidities (Table 4). We categorized cohorts based on age groups and comorbidities including PCOS, obesity, HPL, HRT, and statin use. With respect to age stratification, two methods were adapted. First, we stratified subjects aged 30–69 years into four age-groups, which contains a 10-year span (30–39, 40–49, 50–59, and 60–69). Additionally, we also categorized subjects into two specific age groups, <50 years and ≥50 years. Since the average age of menopause in Taiwanese was 50.2 years [19], we therefore used the age of 50 as the cut-off age of menopause in the study. Accordingly, age <50 years represents pre-menopause, whereas ≥50 years represents post-menopause. We observed that subjects in nearly all categories had higher RR of developing uterine cancer as long as DM was present except in the 60–69 age group and those under HRT. Significantly, the effect of DM on uterine cancer was highest in the age group 30–39 (RR 3.05, 95% CI 2.35–3.96, *p* < 0.0001). DM was positively associated with developing uterine cancer, but RR gradually decreases with aging (Table 4). 

### 3.6. Age and Risk Factors for Uterine Cancer 

Since the risks of uterine cancer development differ with age, menopause, and HRT, we further divided age groups into <50 years (premenopausal) and ≥50 years (postmenopausal) groups for further survey. After using the multivariate Cox proportional hazard regression model and age stratification, the risk of developing uterine cancer was statistically significant in subjects with DM, obesity, and under HRT in both age groups (Table 5). Premenopausal subjects (<50 years) with DM (HR 2.39, 95% CI 2.09–2.74, *p* < 0.0001) were at a significantly higher risk of developing uterine cancer compared to postmenopausal subjects with DM (HR 1.38, 95% CI 1.25–1.53, *p* < 0.0001). The association between HPL and uterine cancer was specifically significant in the <50 years group (HR 1.33, 95% CI 1.22–1.46, *p* < 0.0001). Similarly, the association between statin users and uterine cancer was also only observed in the <50 years group (HR 1.27, 95% CI 1.12–1.44, *p* = 0.0002). In contrast, in the ≥50 years group, we observed a 9% reduction in the risk of uterine cancer (HR 0.91, 95% CI 0.84–0.98, *p* = 0.012) in subjects with HPL, and a 5% reduction in risk in subjects prescribed statins, although it was not significant (HR 0.95, 95% CI 0.88–1.04, *p* = 0.256). However, if we further considered the interactions between HPL and statin use after age stratification and corresponding adjustments, HPL alone, statin use alone, and HPL plus statin use all showed trends of risk reduction in developing uterine cancer in ≥50 years group (Appendix A). In particular, a significant risk reduction was seen in the ≥50 years group with HPL and statin use (adjusted HR 0.85, 95% CI 0.78–0.94, *p* = 0.001) (Appendix A).

Regarding obesity, both age groups had a similar risk of developing uterine cancer (<50 years, HR 2.06, 95% CI 1.75–2.41, *p* < 0.0001; ≥50 years, HR 2.05, 95% CI 1.67–2.51, *p* < 0.0001). HRT was positively associated with uterine cancer in both age groups, but the risk was even higher in the ≥50 years group (<50 years, HR 1.52, 95% CI 1.21–1.90, *p* = 0.0003; ≥50 years, HR 2.10, 95% CI 1.50–2.94, *p* < 0.0001). Since PCOS cannot be diagnosed after menopause, the association between PCOS and uterine cancer was only observed in the premenopausal group (HR 2.03, 95% CI 1.72–2.38, *p* < 0.0001).

### 3.7. Risk of Uterine Cancer Development by DM and Comorbidities Stratification

The numbers, person years, incidence of uterine cancer, and adjusted HR were further calculated in different sub-groups stratified by diabetes and comorbidities simultaneously (Table 6). Since DM, obesity, PCOS, HPL, statin use, and HRT were all associated with uterine cancer development in the age group of <50 years (Table 5), we further attempted to clarify a given risk factor or comorbidity contributing to an increased risk of uterine cancer in the presence of DM or not. We used a stratified analysis with adjustment for each given risk factor or comorbidity. To calculate the specific adjusted HR, HR of the group without a given risk factor or comorbidity was set as a reference (Table 6). After stratification, a subject with PCOS (HR 2.31, 95% CI 1.95–2.74, *p* < 0.0001) was significantly associated with uterine cancer compared to subjects without PCOS in a non-DM group. A similar effect was also observed in the non-DM group with HRT (HR 1.72, 95% CI 1.42–2.08, *p* < 0.0001). Higher HR was found in DM with PCOS (HR: 1.66, 95% CI: 0.92–2.99, *p* = 0.094) or DM with HRT (HR: 1.20, 95% CI: 0.45–3.22, *p* = 0.71) without statistical significance. Obesity is the comorbidity associated with uterine cancer in both non-DM and DM groups (HR 2.19, 95% CI 1.91–2.51, *p* < 0.0001 vs. HR: 1.74, 95% CI: 1.27–2.40, *p* = 0.0006). Although PCOS and HRT were risk factors for uterine cancer, significance was only observed in the non-DM group in this analysis. This result might be attributed to the small sample size in subjects of DM with uterine cancer that was further stratified by PCOS or HRT. 

## 4. Discussion

Uterine cancer is a worldwide public health issue. In the United States, the median age of uterine cancer diagnosis was 63 years from 2012 to 2016 [20]. However, in Taiwan, the age of uterine cancer diagnosis was lower, as the incidence peak of uterine carcinoma was between 50 and 59 years, while between 40 and 49 years of uterine sarcoma from 1979 to 2008 [6]. Since most uterine cancer was diagnosed before age 70, women older than 70 years were excluded in our study. Although the data were extracted from different time frames, our findings extracted from 2005 to 2015 also supported a previous report that the age group of 50–59 years had the highest risk of developing uterine cancer followed by the age group 40–49 years. Both findings suggest a trend of younger diagnosis of uterine cancer in Taiwan. The increase in uterine cancer during the last 30 years was mainly attributed to the increase in type 1 EC [6], and our results also showed that the main subtype of uterine cancer was endometrioid adenocarcinoma. Our finding was consistent with the current global trend of carcinoma as the predominant cancer subtype in uterine cancer. In our report, we found no differences in uterine cancer subtypes between the non-DM and DM groups. This finding suggests that DM per se might not increase the incidence of a given subtype of uterine cancer in Taiwan. A similar finding was also reported in Australia that parity, oral contraceptive use, smoking, age at menarche, and DM were not associated with any given subtype of uterine cancer. However, BMI was specifically associated with type 1 tumors [21]. However, we did not observe any associations between obesity or DM and uterine cancer subtype in Taiwanese populations.

The WHO estimated that the worldwide prevalence of DM was 171 million in 2000, and would be approximately 366 million by 2030 [22]. In Taiwan, the prevalence of DM gradually increased and was 9.82% in subjects aged 18 years and older [23]. From 2005 to 2014, the total population of DM in Taiwan increased by 66%, and the age-standardized prevalence increased by 41% in the population aged 20 to 79 years [24]. The change was largely caused by the increase in the young population [9]. Type 2 DM has been shown to be significantly associated with a higher risk of developing EC [25]. A meta-analysis showed that women with DM had a 72% increased risk of EC compared to those without DM [26]. The highest relative risk was seen especially in women under 50 years of age [13]. In our results, we found that age was positively associated with uterine cancer of all age groups; in particular, the 50–59 age group was associated with the highest risk followed by age groups 40–49 and 60–69. In 2017, an association between type 2 DM and cancer incidence in Taiwan was reported, showing that younger participants (<55 years) with DM had a higher risk of all-cause cancers compared to those in the same age group without DM [27]. Chen et al. in Taiwan also reported a similar result—that DM significantly increased the risk of endometrial cancer in women of all ages, and the highest risk of endometrial cancer was observed in those younger than 50 years [13]. However, they did not adjust for possible confounders such as BMI, obesity, and unopposed estrogen exposure, which may also affect the incidence of uterine cancer. We observed an elevated risk of uterine cancer in women with DM of all ages. After adjusting confounders, the association between DM and uterine cancer remained in all age groups, but the effect (HR) was greater in the <50 years group than the ≥50 years group. Accordingly, DM is the most important risk factor for uterine cancer compared to other metabolic comorbidities in younger patients in our study.

We also found that obesity was associated with an increased risk of uterine cancer in both the DM and non-DM groups. Obese women were at two to four times greater risk of developing uterine cancer than those with normal body weight in the United States [28]. In a recent umbrella review, a positive association was found between high BMI, waist-to-hip ratio, and endometrial carcinomas in premenopausal women [29]. Plausible mechanisms were proposed that obesity increases uterine cancer risk by augmenting aromatase activity, which converts androgen to estrogen to promote endometrial proliferation [30]. Although we and others found that obesity was positively associated with uterine cancer, this result should be interpreted with caution. First, obesity might be underestimated, since the coding of obesity other than morbid obesity tended to be missed in the medical record. Second, obesity was defined by the value of BMI; however, Asians tend to have a lower BMI but a higher percentage of body fat than the white population [31]. In South-East Asia, waist circumference was associated with metabolic abnormalities and is more correlated with visceral adiposity than BMI [32]. Therefore, waist circumference or waist-to-hip ratio might be a more appropriate surrogate than BMI for Asians in the diagnosis of obesity. Thus, the risk of obesity on uterine cancer might be underestimated in our population.

In addition to the associations between DM, obesity, and uterine cancer development, there was also increased cancer mortality in patients with DM after age and comorbidities were adjusted [33]. This phenomenon could be explained by a variety of metabolic abnormalities including hyperinsulinemia, insulin resistance, increased insulin-like growth factor-1 (IGF-1) level, dyslipidemia, augmented inflammatory cytokines, elevated leptin, as well as decreased adiponectin [33]. These metabolic abnormalities are commonly seen in PCOS, which is a common endocrine disorder in young women. A meta-analysis showed that women with PCOS were at a three times higher risk of developing endometrial cancer [34]. In our study, the diagnosis of PCOS was not different between the DM and non-DM groups; however, PCOS was significantly associated with uterine cancer after age, DM, obesity, HPL, statin use, and HRT were adjusted. However, this association was only found in subjects <50 years since PCOS is a premenopausal disease. Specifically, the presence of PCOS would further increase risk of uterine cancer; however, the effect was modest in subjects with DM. This might suggest that PCOS also behaves as a risk factor, but it was overshadowed by DM if both diagnoses were present in analysis. Alternatively, it can be explained that the small sample sizes in the DM subjects with PCOS render the non-significance. The effects of PCOS on uterine cancer might be explained by the aforementioned metabolic abnormalities PCOS exhibits that lead to an unopposed estrogen and increased risk of uterine cancer [35].

In our cohort, nearly 40% of the subjects in the DM group had HPL, whereas only 8% in the non-DM group. A case-control study suggested that lower levels of serum cholesterol and low-density lipoprotein cholesterol (LDL) were associated with a higher risk of endometrial cancer [36]. However, no association of total cholesterol, LDL cholesterol, and triglyceride (TG) levels with the risk of endometrial cancer was found after the adjustment of BMI [37]. A Norway study in 2009 found that only serum TG level, but not total cholesterol, LDL, or high-density lipoprotein cholesterol (HDL), was positively associated with the risk of endometrial cancer [12]. However, we found that HPL was associated with uterine cancer specifically in the age group <50 years after adjustment. Similar to HPL, statin use was associated with a greater risk of uterine cancer in the <50 years age group. Since statins were prescribed in HPL, this phenomenon of statins was possibly driven by HPL itself. Interestingly, in those with an age ≥50 years, HPL and statin use were associated with a reduced risk of uterine cancer. The statin use in the post-menopause group seems protective against uterine cancer even though HPL was not present. This age-specific effect of statins was not previously mentioned [38]. A meta-analysis also found that statin use was associated with a suggestive reduction in the risk of uterine cancer in Asian populations [38]. Lavie et al. reported that statins are associated with decreased cancer risk and improved survival in endometrial cancer by blocking cancer cell growth pathways and induction of tumor cell apoptosis [39]. Currently, the age-specific risk of HPL and uterine cancer remains unclear. Our findings suggest HPL and statin use might act as either a risk factor or a protective factor of uterine cancer according to specific ages.

In our study, HRT was positively associated with uterine cancer development after age, DM, and other comorbidities were adjusted. No significance was found in DM subjects with HRT and uterine cancer, which could be explained by the small sample size of the DM with HRT group. Different regimens of HRT (estrogen only or estrogen plus progestin) might also affect cancer risk. In subjects diagnosed with uterine cancer, HRT was associated with higher cancer risk in both premenopausal and postmenopausal groups. Endometrial cancer was strongly associated with the use of unopposed estrogen in postmenopausal women without hysterectomies [40]. This risk, however, can be mitigated if progestins were added. However, it remains debatable whether continuous combined therapy provides a greater protection of the uterus than placebo or never-use. A review concluded that the use of estrogen alone, tibolone, and sequential combined therapy could increase the risk of endometrial cancer, even when treatment lasts less than five years, but continuous combined therapy could present a lower risk than never use even for more than 10 years [41].

There were some strengths as well as limitations of our study. Regarding the strengths, the study cohort was collected from the NHI database with links to the TCR database, which is a representative nationwide database with a large sample size and high completeness. Second, multivariate Cox proportional hazard regression models with adjustment for potential confounding factors were performed. Third, age-stratification was carried out to analyze the risk in premenopausal and postmenopausal populations. There are some limitations in this study; first, we cannot differentiate type 1 or type 2 DM in our cohort, but we excluded DM subjects younger than 30 years old to ensure that most of the DM were type 2. Second, information on the duration or severity of DM and treatment regimens was lacking in our cohort. Studies have reported that a significantly increased incidence of uterine cancer was seen within the first month after type 2 DM was diagnosed [42], or even in the pre-diabetes phase [43,44]. However, a recent study suggested that the highest risk of uterine cancer was 4 to 8 years after the diagnosis of type 2 DM and remains at a plateau afterward [45]. Third, information of BMI, waist-to-hip ratio, or body weight to determine the degree of obesity, potential confounders including parity, age of menarche, or menopause were not available. Fourth, treatment duration of HRT and statin use or some given medications such as metformin cannot be extracted from the database.

## 5. Conclusions

In this nationwide population-based study, we have identified positive associations between DM and uterine cancer of all ages, and this risk was much higher in the younger subjects, especially in the age group 30–39 years. Similarly, obesity, PCOS, HPL, statin use, and HRT were also significantly associated with uterine cancer in subjects younger than 50 years of age. Among all risk factors of uterine cancer, DM is overall the most important one. We instead observed a protective effect of HPL and statin use on uterine cancer at an age ≥50 years. Therefore, subjects with DM, especially younger women with respective comorbidities, should be recommended to have appropriate uterine cancer screenings to achieve early detection. Furthermore, HRT should be prescribed with caution, especially in postmenopausal women with DM or obesity.

## Figures and Tables

**Figure 1 cancers-14-04912-f001:**
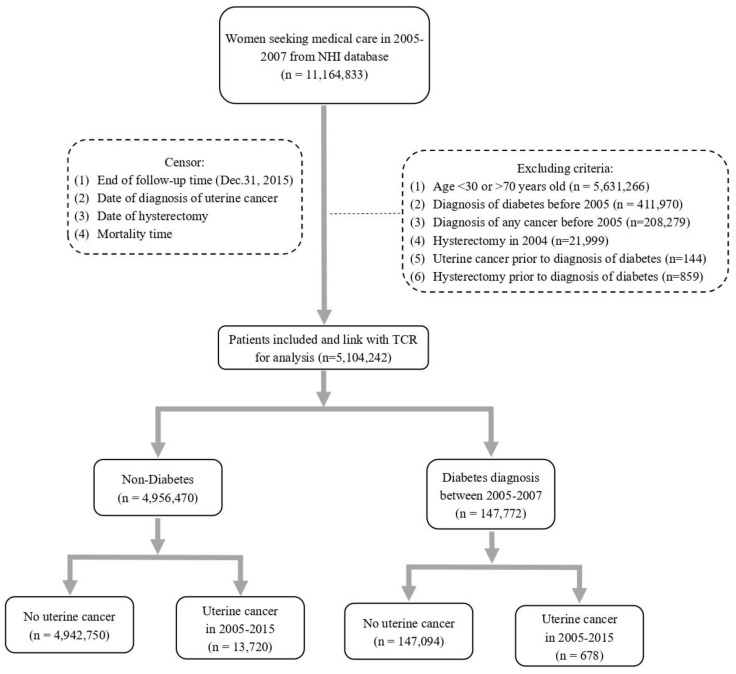
Flow chart for selecting study subjects.

**Figure 2 cancers-14-04912-f002:**
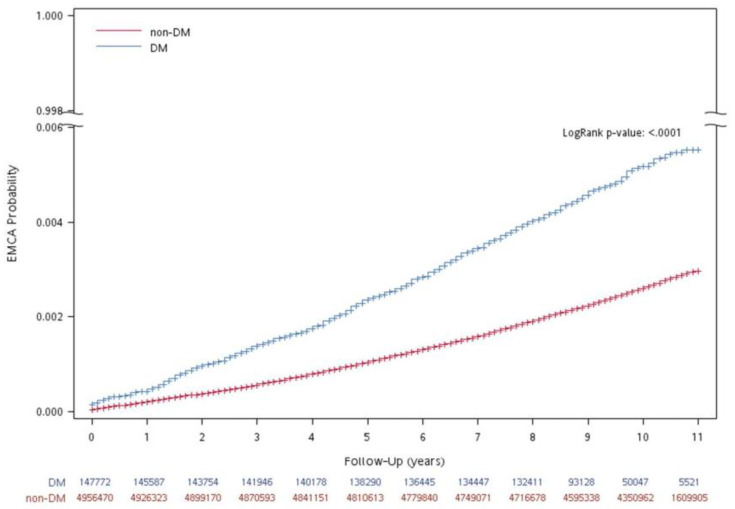
Accumulated probability of uterine cancer in participants with and without DM during follow-up years.

**Table 1 cancers-14-04912-t001:** Basic characteristics of the study population at baseline (*n* = 5,104,242).

Variables	Non-DM		DM		*p*
	*n*	%	*n*	%	
All	4,956,470	97.10	147,772	2.9	
**Age (years) (mean ± SD)**	45.13 ± 10.19		54.55 ± 9.20		<0.0001
**Age group**					<0.0001
30–39	1,718,942	34.68	10,374	7.02	
40–49	1,643,309	33.15	30,770	20.82	
50–59	1,032,226	20.83	57,546	38.94	
60–69	561,993	11.34	49,082	33.21	
**Comorbidity**					
PCOS	26,707	0.54	844	0.57	0.0947
Obesity	34,265	0.69	4566	3.09	<0.0001
HPL	409,784	8.27	58,252	39.42	<0.0001
HRT	23,558	0.48	649	0.44	0.0465
**Statin use**	262,956	5.31	61,333	41.51	<0.0001

DM denotes diabetes mellitus, PCOS polycystic ovarian syndrome, HPL hyperlipidemia, HRT hormone replacement therapy.

**Table 2 cancers-14-04912-t002:** (**A**). Basic characteristics of subjects diagnosed with uterine cancer by DM (*n* = 14,398), (**B**). Subtypes of uterine cancer in the study populations (*n* = 14,398).

Variables	Non-DM		DM		*p*
	*n*	%	*n*	%	
Uterine cancer	13,720	0.28	678	0.46	<0.0001 *
A. Basic characteristics of subjects diagnosed with uterine cancer by DM (*n* = 14,398)					
**Age (years) (mean ± SD)**	47.62 ± 8.56		51.79 ± 8.65		<0.0001
**Age group**					<0.0001
30–39	2456	17.90	63	9.29	
40–49	5740	41.84	185	27.29	
50–59	4196	30.58	296	43.66	
60–69	1328	9.68	134	19.76	
**Comorbidity**					
PCOS	138	1.01	12	1.77	0.0558
Obesity	211	1.54	42	6.19	<0.0001
HPL	1405	10.24	242	35.69	<0.0001
HRT	106	0.77	4	0.59	0.594
**Statin use**	919	6.70	259	38.20	<0.0001
B. Subtypes of uterine cancer in the study populations (*n* = 14,398)					
**Subtype of uterine cancer**					0.0329 ^#^
**Carcinoma**	8177	59.60	406	59.88	0.3311
Endometrioid	7667	55.88	377	55.60	
Clear cell	149	1.09	5	0.74	
Serous	8	0.06	0	0	
Carcinosarcoma	353	2.57	24	3.54	
**Sarcoma**	655	4.77	18	2.65	0.7085
ESS	98	0.71	3	0.44	
LGESS	153	1.12	3	0.44	
LMS	343	2.5	9	1.33	
Adenosarcoma	61	0.44	3	0.44	
**Others**	4888	35.63	254	37.46	

DM denotes diabetes mellitus, PCOS polycystic ovarian syndrome, HPL hyperlipidemia, HRT hormone replacement therapy, * DM & Uterine cancer two by two table chisquare *p* value, # Chi-square test for comparison between three subtypes of uterine cancer. DM denotes diabetes mellitus, ESS endometrial stromal sarcoma, LGESS low-grade endometrial stromal sarcoma, LMS leiomyosarcoma.

**Table 3 cancers-14-04912-t003:** Hazard ratio (HR) of incidence of uterine cancer by Cox proportional hazard regression model.

Variables	Univariate			Multivariate		
	HR	95% CI	*p*	HR	95% CI	*p*
**Age group, years**						
**30–39**	1					
40–49	2.49	2.38–2.61	<0.0001	2.53	2.41–2.65	<0.0001
50–59	2.87	2.73–3.01	<0.0001	2.87	2.72–3.01	<0.0001
60–69	1.72	1.61–1.83	<0.0001	1.69	1.58–1.81	<0.0001
**No DM**	1					
**DM**	1.99	1.85–2.15	<0.0001	1.66	1.53–1.81	<0.0001
**Comorbidity**						
PCOS (vs. no PCOS)	1.90	1.62–2.23	<0.0001	2.91	2.47–3.42	<0.0001
Obesity (vs. no Obesity)	2.36	2.08–2.67	<0.0001	2.13	1.88–2.41	<0.0001
HPL (vs. no HPL)	1.28	1.22–1.35	<0.0001	1.01	0.96–1.07	0.6761
HRT (vs. no HRT)	1.61	1.34–1.94	<0.0001	1.60	1.33–1.93	<0.0001
**Statin use (vs. no statin use)**	1.33	1.25–1.41	<0.0001	1.01	0.94–1.08	0.8065

Multivariate with adjustment for age, DM, Statin use, PCOS, Obesity, hyperlipidemia and HRT, Ref denotes reference, DM diabetes mellitus, HR hazard ratio, HPL hyperlipidemia, CI confidence interval, PCOS polycystic ovarian syndrome, HRT hormone replacement therapy.

**Table 4 cancers-14-04912-t004:** Relative Risk (RR) of developing uterine cancer in subjects with DM versus without DM in different groups.

Variables	RR	95% CI	*p*
**Age group (years)**			
30–39	3.05	2.35–3.96	<0.0001
40–49	1.58	1.36–1.83	<0.0001
50–59	1.28	1.13–1.44	<0.0001
60–69	1.12	0.94–1.34	0.2115
**Age group (years)**			
<50	1.80	1.58–2.05	<0.0001
≥50	1.23	1.11–1.35	<0.0001
**Comorbidity**			
PCOS			
No	1.37	1.27–1.49	<0.0001
Yes	2.74	1.49–5.03	0.0012
Obesity			
No	1.38	1.28–1.50	<0.0001
Yes	1.49	1.06–2.09	0.0201
HPL			
No	1.49	1.36–1.64	<0.0001
Yes	1.23	1.07–1.41	0.0035
HRT			
No	1.39	1.28–1.51	<0.0001
Yes	0.92	0.33–2.54	0.866
**Statin use**			
No	1.48	1.34–1.64	<0.0001
Yes	1.20	1.05–1.38	0.0095

Adjust variables: Age, PCOS, Obesity, HPL, HRT, RR denotes relative risk, DM diabetes mellitus, CI confidence interval, PCOS polycystic ovarian syndrome, HPL hyperlipidemia, HRT hormone replacement therapy.

**Table 5 cancers-14-04912-t005:** Hazard Ratio (HR) of incidence of uterine cancer in age < 50 years and ≥50 years by Cox proportional hazard regression model.

Variables	Univariate			Multivariate		
	HR	95% CI	*p*	HR	95% CI	*p*
**Uterine cancer < 50 years old**						
**DM (vs. no DM)**	3.03	2.67–3.44	<0.0001	2.39	2.09–2.74	<0.0001
**Comorbidity**						
PCOS (vs. no PCOS)	2.19	1.86–2.57	<0.0001	2.03	1.72–2.38	<0.0001
Obesity (vs. no Obesity)	2.56	2.18–2.99	<0.0001	2.06	1.75–2.41	<0.0001
HPL (vs. no HPL)	1.61	1.49–1.75	<0.0001	1.33	1.22–1.46	<0.0001
HRT (vs. no HRT)	1.57	1.25–1.96	<0.0001	1.52	1.21–1.90	0.0003
**Statin use (vs. no statin use)**	1.84	1.65–2.05	<0.0001	1.27	1.12–1.44	0.0002
**Uterine cancer ≥ 50 years old**						
**DM (vs. no DM)**	1.35	1.22–1.49	<0.0001	1.38	1.25–1.53	<0.0001
**Comorbidity**						
PCOS (vs. no PCOS)	-	-	-	-	-	-
Obesity (vs. no Obesity)	2.07	1.69–2.53	<0.0001	2.05	1.67–2.51	<0.0001
HPL (vs. no HPL)	0.93	0.87–1.00	0.0435	0.91	0.84–0.98	0.0122
HRT (vs. no HRT)	2.10	1.50–2.94	<0.0001	2.10	1.50–2.94	<0.0001
**Statin use (vs. no statin use)**	0.97	0.90–1.04	0.4012	0.95	0.88–1.04	0.2558

Multivariate with adjustment for age, DM, Statin use, PCOS, Obesity, HPL, and HRT, HR denotes hazard ratio, CI confidence interval, DM diabetes mellitus, PCOS polycystic ovarian syndrome, HPL hyperlipidemia, HRT hormone replacement therapy.

**Table 6 cancers-14-04912-t006:** Numbers, person years, incidence of uterine cancer, and adjusted HR in different subgroups stratified by diabetes and comorbidities.

		Total Populations		Uterine Cancer		Adjusted HR	95% CI	*p*
		*N*	Person-Years	*N*	Person-Years			
Non-DM	without PCOS	4,929,763	51,171,301.2	13,582	82,886.8	Ref	Ref	-
	with PCOS	26,707	282,216.3	138	844.9	2.31	1.95–2.74	<0.0001
DM	without PCOS	146,928	1,326,044.5	666	3370.2	Ref	Ref	-
	with PCOS	844	7741.3	12	46.8	1.66	0.92–2.99	0.0943
Non-DM	without HRT	4,932,912	51,208,704.6	13,614	83,155	Ref	Ref	-
	with HRT	23,558	244,812.9	106	576.7	1.72	1.42–2.08	<0.0001
DM	without HRT	147,123	1,328,065.8	674	3398.5	Ref	Ref	-
	with HRT	649	5720	4	18.5	1.20	0.45–3.22	0.7135
Non-DM	without HPL	4,546,686	47,149,557.9	12315	75237.6	Ref	Ref	-
	with HPL	409,784	4,303,959.6	1405	8494.1	1.01	0.95–1.07	0.7317
DM	without HPL	89,520	797,265.8	436	2154.1	Ref	Ref	-
	with HPL	58,252	536,520	242	1262.9	0.86	0.74–1.01	0.0633
Non-DM	without Obesity	4,922,205	51,097,078.6	13,509	82,483	Ref	Ref	-
	with Obesity	34,265	356,438.9	211	1248.7	2.19	1.91–2.51	<0.0001
DM	without Obesity	143,206	1,292,094.3	636	3217	Ref	Ref	-
	with Obesity	4566	41,691.5	42	200	1.74	1.27–2.40	0.0006

Adjust variables: Age, PCOS, Obesity, HPL, HRT, Ref denotes reference, HR hazard ratio, CI confidence interval, DM diabetes mellitus, PCOS polycystic ovarian syndrome, HRT hormone replacement therapy, HPL hyperlipidemia.

## Data Availability

The datasets generated and/or analyzed during the current study are not publicly available in accordance with the policy of the Health and Welfare Data Science Center, Ministry of Health and Welfare, Taiwan, but are available from the corresponding author upon reasonable request.

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
