# Peer review of "Age Specific Risks of Uterine Cancer in Type 2 Diabetes and Associated Comorbidities in Taiwan"

_cancers, 2022, doi:10.3390/cancers14194912_

Round 1

Reviewer 1 Report

well written

Author Response

Thank you for your comment.

Reviewer 2 Report

Thank you very much for giving me the opportunity to review this article. This article is well written. The sample size is large and is representative. 

There are several comments:

1. Why over 70 years old were excluded? Could the authors give reasons on this in the discussion part?

2. Is there any data on the use of DM medications? Any data on metformin, insulin or SGLT2 inhibitors? Any correlations with the incidence of uterine cancer?

3. Is there any data on hypertension? It would be more informative to have correlation with metabolic syndrome (hyperlipidemia, HT, DM)

Overall the article is well written and informative.

Author Response

Dear reviewer, 

Reviewer 3 Report

This study studied the risk factor for uterine cancer in a large cohort with more than 5 millions of human subjects. With the 11 yeard of follow-up study, the study confirmed type 2 diabetes as a major risk factor of uterine cancer. This study certainly presented interesting observations with important clinical implications. I would recommend the immediate acceptance of this well-written manuscript. 
